# Effect of Preparation Parameters on Microparticles with High Loading Capacity and Adsorption Property Adsorbed on Functional Paper

**Zuobing Xiao, Shuai Wan, Yunwei Niu and Xingran Kou \***

School of Perfume and Aroma Technology, Shanghai Institute of Technology, Shanghai 201418, China; xzb@sit.edu.cn (Z.X.); 176071228@mail.sit.edu.cn (S.W.); nyw@sit.edu.cn (Y.N.)
\* Correspondence: kouxr@sit.edu.cn

**Abstract:** Microparticles encapsulated with orange essential oil were prepared by improved emulsifying solvent volatilization technology, and modified with chitosan to improve their loading and adhesion properties on paper. Characterization was performed by Zetasizer Nano ZS instrument, transmission electron microscope (TEM), scanning electron microscopy (SEM), Fourier transform infrared spectrometer (FTIR) spectroscopy, thermogravimetric analyzer (TGA), gas-chromatography-mass spectrometry (GC-MS) and the ultrafast GC Electronic Nose Heracles II, etc. The results showed that for poly (lactic-co-glycolic acid) (PLGA) microparticles and chitosan–PLGA microparticles, respectively, the particle sizes were 233.2 and 277.6 nm, loading capacity was 19.17% and 24.36%, Zeta potential was −8.27 and 5.44 mV, adhesive capacity was 76.32 and 324.84 mg/g, and encapsulation efficiency was 93.23% and 94.06%. GC-MS demonstrated that the embedding process minimally effected the aroma quality of orange essential oil. The ultrafast GC Electronic Nose Heracles II showed that chitosan–PLGA microparticles could effectively slow the release of the orange essential oil. Therefore, this work provides a proposal for a better understanding of biodegradable functional packaging paper.

**Keywords:** PLGA; chitosan; microparticles; slow-release; adhesive capacity; functional paper

## 1. Introduction

Essential oils, which are composed of various volatile aromatic substances, are extracted from flowers, leaves, stems, roots or fruits of plants by steam distillation, extrusion, cold immersion or solvent extraction. In recent years, essential oils as natural products have been widely used in packaging coatings [1,2], especially in the field of active packaging [3]. These packaging coatings are particularly prominent for their antibacterial and anticorrosive performances. Orange essential oil, which is widely used in food and medicinal industries is obtained by pressing or steam distillation from the peel of sweet orange. However, essential oils are susceptible to environmental factors, such as temperature, pH, humidity and light [4]. These can cause essential oils to deteriorate, which affects their quality and application properties. Previous studies have shown that microencapsulation is an excellent way to reduce the influence of environmental factors on essential oil [5,6]. Currently, microcapsule products are used in many fields, such as toiletries, food, medicine, household chemicals and biological research. Mostly, the microencapsulation materials are non-degradable, which inevitable harm the environment. For instance, silver nanoparticles, $SiO_2$ nanoparticles and $TiO_2$ nanoparticles are widely used in biomedicines and foods [7–9]. Once absorbed by humans or other organisms through breathing, these nanoparticles are likely to cause lung damage [10]. Interestingly, biodegradable block copolymers have shown obvious advantages as embedding materials in recent years.

PLGA is a copolymer of lactic acid (LA) and glycolic acid (GA) by polymerization reaction [11]. Because it is biodegradable [12] and safe, the Food and Drug Administration has given permission for PLGA to be used in drug delivery systems [13], and it is also recorded in the US Pharmacopeia [14]. Zhu prepared PLGA nanoparticles with sizes of 150 nm to encapsulate drugs, and indicated that the nanoparticles exhibited an excellent slow-release feature as well as high loading capacity and safety [15,16]. Therefore, PLGA is considered as an excellent coating material for biomedicine and packaging materials.

However, making PLGA microparticles adsorb to a paper surface is difficult because of electrostatic repulsion between the hydroxyl and carboxyl groups on paper fibers and the terminal carboxyl on PLGA microparticles. Therefore, to modify the PLGA microparticles is necessary, so that their adhesive capacity can be improved. Chitosan, as a biodegradable material [17], is widely used in antibacterial packaging coatings [18,19]. The abundant amino groups on chitosan can combine with the hydroxyl and carboxyl groups on paper fibers [20]. Therefore, using chitosan to modify the PLGA microparticles may improve the adhesion property of microparticles on paper surfaces.

In the present work, PLGA was used as a wall material to prepare microparticles. The biodegradable polymer chitosan was used to modify the surface of PLGA microparticles to improve their loading capacity and adsorption property on paper surface. The adsorption capacity and sustained-release properties of the functional paper were studied.

## 2. Materials and Methods

### 2.1. Materials

The orange essential oil was purchased from Apple Flavor & Fragrance Group Co., Ltd. (Shanghai, China). Its composition was analyzed by GC-MS (GCMS-QP 2010 Ultra, Tsushima, Japan). PLGA (Mw 10,000 Da), was purchased from Jinan Jufukai Biotechnology Co., Ltd. (Jinan, China). Polyvinyl alcohol (PVA, Mw 74,800 Da) was purchased from Shanghai Titan Technology Co., Ltd. (Shanghai, China). Ethyl acetate was purchased from Shanghai Titan Technology Co., Ltd. (Shanghai, China). Chitosan (Mw 260 kDa) with the deacetylation degree of 95% was purchased from Shanghai Titan Technology Co., Ltd. (Shanghai, China). Distilled water was prepared in our laboratory. Ethanol (Anhydrous ≥99.7%, AR) was purchased from Shanghai Titan Technology Co., Ltd. (Shanghai, China). Potassium bromide (AR, Anhydrous ≥99%) was purchased from Shanghai Titan Technology Co., Ltd. (Shanghai, China). All chemicals were of analytical grade.

### 2.2. Preparation of Microparticles

The process for preparing water soluble microparticles by a modified emulsion solvent evaporation technology was as follows [21,22]. Briefly, the organic phase consisted of 100 mg of PLGA and 100 mg orange essential oil which were previously dissolved in 5 mL of ethyl acetate. PVA (different concentrations 0.5%, 1%, 1.5%, 2%, and 2.5% w/v) was dissolved in 40 mL deionized water to form the aqueous phase. The organic phase was injected into the aqueous phase at 25 °C with a controlled flow rate using an LSP syringe pump (LSP01-1A, Baoding Dichuang electronic technology co. Ltd., Baoding, China) at 2 mL/min. After the organic phase was completely injected, the mixture was stirred for 30 min. The agitation rate was produced and controlled by constant temperature magnetic stirrer (524G, Mei yingpu instrument manufacturing co. LTD, Shanghai, China). The mixed solution was dispersed by ultrasonication (100 W, 5 min) in an ice bath. It was then stirred for 1 h at 40 °C, the organic phase and part of the aqueous phase were evaporated by vacuum distillation, and the system became a uniform translucent emulsion with blue opalescence.

### 2.3. Modification of Microparticles with Chitosan

Chitosan was used to modify the PLGA microparticles according to a previous study [23]. Briefly, the aqueous phase was 1% PVA and 0.2% chitosan in 1% glacial acetic acid solution. The organic

phase (prepared as described in Section 2.2) was injected into the aqueous phase at room temperature with a controlled flow rate of 2 mL/min and an agitation rate of 800 rpm. After the organic phase was completely injected, the mixture was stirred for 1 h at 25 °C. The organic phase and part of the aqueous phase were then evaporated by vacuum distillation, and the system became a uniform translucent emulsion with blue opalescence. The microparticles system was stored at room temperature (25 °C) for further analysis and application.

### 2.3.1. Particle Size, Zeta Potential and Surface Morphology of Microparticles

The microparticles were sufficiently dispersed after diluting 10 times with distilled water, which was used to characterizing the particle size, zeta potential and surface morphology. The microparticles of particle size, zeta potential and surface morphology were analyzed using dynamic light scattering on a Zetasizer Nano ZS instrument (ZEN3600, Malvern, UK) to determine the particle size and zeta potential. The temperature was maintained at 25 °C, and the scattering angle was 173° from the incident laser beam. The appearance of the microparticles was observed using a transmission electron microscope (TEM, Tecnai G2 F20 TWIN, FEI, Hillsboro, OR, USA) with an accelerating voltage of up to 200 kV. A total of 5 µL of sample solution was placed on copper grids with a continuous carbon film coating, then the solvent was evaporated at 25 °C.

### 2.3.2. Fourier Transform Infrared (FTIR) Spectroscopy

FTIR (Nicolet iN10, Thermo Fisher Scientific, Waltham, MA, USA) was employed to determine whether the orange essential oil was encapsulated in the PLGA or chitosan–PLGA microparticles by measuring the characteristic peaks of different groups. The PLGA, PLGA microparticles and chitosan–PLGA microparticles were mixed with KBr to form transparent sheets. The resulting sheets were characterized by FTIR, within the wavenumber range 400–4000 $cm^{-1}$.

### 2.3.3. Loading Capacity and Encapsulation Efficiency of Microparticles

Full-wavelength scanning of the orange essential oil was performed to determine the maximum absorption wavelength, which was 260 nm. Five different concentrations of orange essential oil were diluted with ethanol, and the absorbance was measured using an ultraviolet spectrophotometer (Alpha-1860S, Puyuan Instrument Co., Ltd. Shanghai). The standard curve was obtained as $Y = 0.3884X + 0.0572$ (where $X$ is the concentration of essential oil, and $Y$ is absorbance); the correlation coefficient of the linear equation of the experimental point regression $R^2$ was 0.992 with excellent linearity.

The loading capacity and encapsulation efficiency of the microparticles were calculated as follows: A sample dried in a freeze dryer (FD-1A-50, Beilang Instrument Co., Ltd. (Shanghai, China)) with weight of W, was added to a certain volume of deionized water for ultrasonic extraction, which was performed in an ultrasonic cleaner tank (300 mm × 240 mm × 150 mm) (SB–5200DTN, SCIENTZ, Ningbo, China) with 40 W for 8 min at 25 °C. Then, the sample was centrifuged at 15,000 rpm, 25 °C for 30 min using an ultracentrifuge (CT15RT, Tinmei (holding) Co., Ltd. (Shanghai, China)), and filtrated using a 0.2 µm filter membrane. The supernatant was collected, and its absorbance was measured with an ultraviolet spectrophotometer. The concentration of the orange essential oil was calculated according to the standard curve. The encapsulation efficiency and loading capacity of microparticles were calculated using Formulas ((1) and (2)).

$$\text{Encapsulation efficiency} = \frac{\text{Mass of total oil} - \text{Mass of free oil}}{\text{Mass of total oil}} \times 100\% \tag{1}$$

$$\text{Loading capacity} = \frac{\text{Mass of total oil} - \text{Mass of free oil}}{\text{Mass of microparticles}} \times 100\% \tag{2}$$

*2.4. Preparation of the Functional Paper*

Three pieces of kraft paper (3 cm × 3 cm) were completely immersed in a 20 mL solution of orange essential oil, PLGA microparticles and chitosan–PLGA microparticles for 30 min, respectively. The wet paper was removed and placed in a ventilated place to dry naturally for 12 h. Finally, the resultant functional papers were placed into the sample bottles of 20 mL and used for the measurement of slow-release study.

*2.5. Adsorption Capacity of Microparticles on Paper*

The adsorption capacity is an essential property of the microparticles on the functional paper. In this work, scanning electron microscope (SEM, S-3400N, Hitachi High-Technologies, Tokyo, Japan) at an acceleration voltage of 15 KV energy with magnifications of 2000 times was used to observe difference in the amount of microparticles with or without modification on the surface of the paper, and the blank paper was magnified 300 times to observe the structure of the paper fiber. Before obtaining images, the samples were coated with gold using a gold sputter coater in a high vacuum evaporator. Thermogravimetric analysis (TGA, Q5000 IR, Waters Corporation, Milford, MA, USA) was performed to quantitatively reveal the adsorption capacity. Based on the weight losses from untreated paper, PLGA microparticles and (chitosan–)PLGA microparticle functional paper from 100 to 500 °C, the adsorption capacity of the functional paper was determined (formula (3)) [24].

$$W_1 \times \left(1 - \frac{q}{1000}\right) + W_2 \times \frac{q}{1000} = f \tag{3}$$

where q (mg/g) is the amount of microparticles adhesive on functional paper, $W_1$ is the quality loss rate from 100 to 500 °C of untreated paper, $W_2$ is the weight loss rate from 100 to 500 °C of microparticles, and f is the weight loss rate for the functional paper from 100 to 500 °C.

*2.6. Statistical Analyses*

All measurements were performed in triplicate. Significant differences among the treatments were analyzed by one-way analysis of variance (ANOVA), and Duncan's post hoc multiple comparisons were performed following one-way ANOVA based on SPSS 21.0 statistical calculations. The statistical significance level was set at $p < 0.05$.

## 3. Results and Discussion

*3.1. Factors of Affecting the Preparation of Microparticles*

3.1.1. Effect of PLGA Type on the Particle Size and the Loading Capacity of Microparticles

In this work, PLGA with different monomer ratios of 75:25 and 50:50 was used as the wall material for microparticles. The effect of monomers with different contents on the particle size and loading capacity of microparticles were compared. There are significant differences between experimental results of different PLGA types in Figure 1A.

The size of the microparticles varied with changes in the self-assembly morphology. Generally speaking, the mass fraction of the hydrophilic chain is higher, and it is easier to form spherical particles; the larger the molecular weight of the block copolymer, the more easily it forms complex structures [25]. In this experiment, the particle sizes of PLGA (75:25) and PLGA (50:50) were 233.2 and 246.9 nm, the PDI values were 0.066 and 0.069, and the loading capacities were 24.35% and 23.85%, respectively. It was clear that the particle size and loading capacity of the microparticles were influenced by the chemical composition of the copolymer. This showed that the size of the microparticles increased with the number of hydrophobic chains increased [26]. Since the copolymer composition of PLGA played an essential role in the particle size and loading capacity of microparticles, the PLGA ratio of 75:25 was selected for further study.

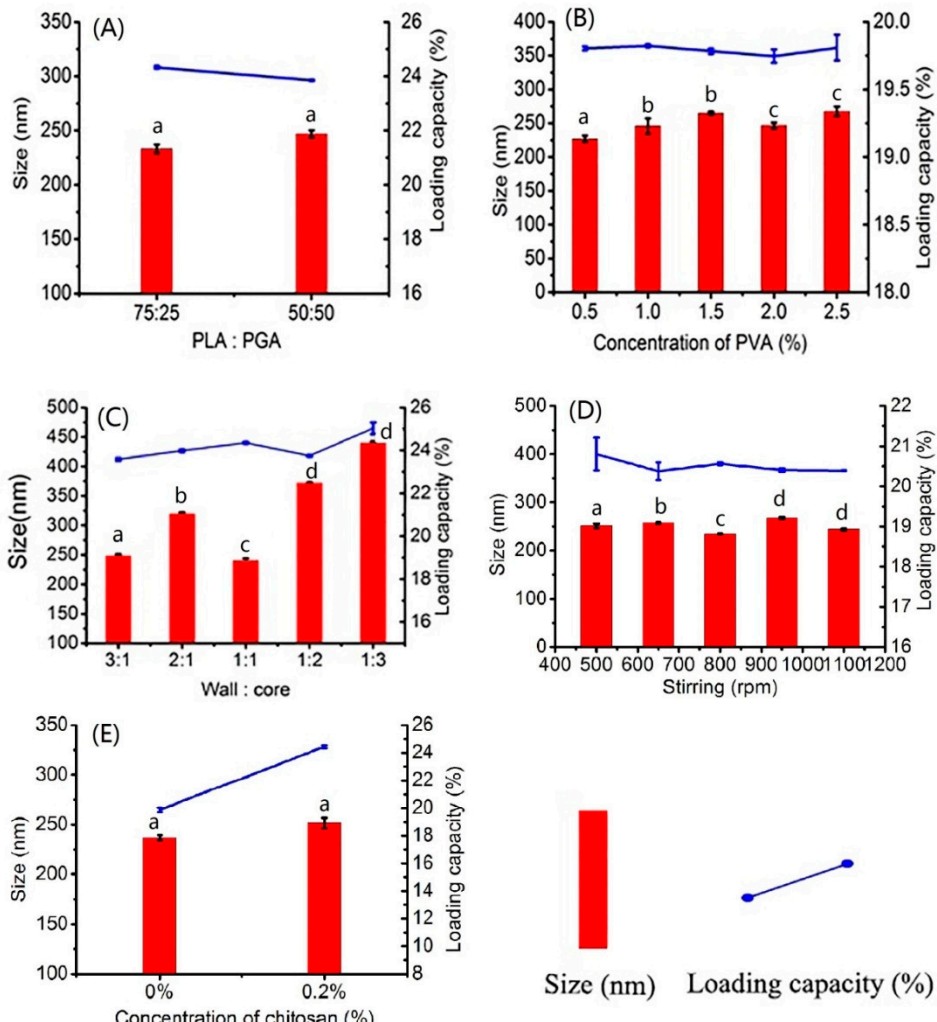

**Figure 1.** (**A**): effect of PLGA (PLA:PGA) on the particle size and the loading capacity, (**B**): effect of PVA concentration on the particle size and the loading capacity, (**C**): effect of wall to core on the particle size and the loading capacity, (**D**): effect of stirring on the particle size and the loading capacity, (**E**): effect of chitosan concentration on the particle size and the loading capacity. Different lowercase letters indicate a significant difference by one-way analysis of variance (ANOVA) ($p < 0.05$).

### 3.1.2. Effect of PVA Concentration on the Particle Size and Loading Capacity of Microparticles

As a dispersing agent, PVA has good water solubility, stability, and safety. It is widely used in the chemical and biomedical industries [27]. In this experiment, the effects of PVA on the particle size and loading capacity were studied by changing the concentrations of PVA. There are significant differences between the experimental results of different PVA concentrations in Figure 1B.

The emulsifier mechanism of PVA can be summarized as steric hindrance and electrostatic stabilization [27], its concentration significantly influenced the dispersion effect of the system. According to the experimental results, the sizes of the microparticles prepared with different concentrations of PVA were 223.5–267.6 nm. When the PVA concentration was 0.5%, two peaks were observed, however, the other concentrations all resulted in single peaks on the particle size distribution maps. That was mainly because the PVA concentration was too small to completely separate from the microparticles. As the PVA concentration increased, the particle size of the microparticles first decreased and then increased. At this low concentration of PVA, the electrostatic attraction between the microparticles was higher than the dispersion caused by PVA. When the PVA concentration was 1.5%, the microparticles were separated, so the particle size decreased. However, beyond this concentration,

the particle size of the microparticles increased again, which was consistent with the results of previous literature [28]. The main reason for this might be that as the viscosity of the aqueous phase increases with the PVA concentration, the less the wavy movement speed probability of the tiny particles are [29]. The sheer force produced by stirring was insufficient to completely separate the microparticles.

### 3.1.3. Effects of Wall Material to Core Material on Particle Size and Loading Capacity of Microparticles

In this experiment, the proportion of wall to core was 3:1, 2:1, 1:1, 1:2, and 1:3, respectively. There was a significant effect of the experimental result in Figure 1C. The experimental result showed that with the increasing of the ratio of wall to core, the particle size of microparticles showed an upward trend, while the loading capacity initially increased and then reached a plateau. At the ratios of 3:1, 2:1, and 1:1, the loading capacity changed slightly. This was mainly because when the amount of essential oil was fixed, the encapsulation efficiency of the sweet orange oil could be influenced by the amount of wall material. It is well known that the ratio of wall material to core material may affect the cost of the final products [30]. Therefore, the best choice of wall material to core material was 1:1.

### 3.1.4. Effects of Stirring Rate on Particle Size and Loading Capacity of Microparticles

During the preparation of microparticles, the stirring rate strongly influenced the particle size and loading capacity. In this experiment, the effects of stirring rate on the particle size and loading capacity of microparticles were investigated with various stirring rates (500–1200 rpm) produced and controlled with a constant temperature magnetic stirrer (524G, Mei Yingpu Instrument Manufacturing Co. Ltd., Shanghai, China).

The results (Figure 1D) showed that as the stirring rate increased, the particle size of the microparticles first decreased and then increased to a plateau. The main reason was that different stirring rates produced different shearing forces, resulting in different particle sizes. Moreover, the nanoparticles size potentiometer showed that when the stirring rate exceeded 800 rpm, the particle size did not change significantly. While the stirring speed was 800 rpm, the size of microparticles was smaller, which was consistent with the previous literature [31]. The main reason for this might be an increase in the shear force caused by increased agitation, causing the particles to be separated better. However, as the stirring rate increased again, the loading capacity changed slightly. Therefore, to save energy, 800 rpm was selected as the stirring speed.

### 3.2. Chitosan Modification of Microparticles

Chitosan, a biodegradable polysaccharide, has excellent film-forming property [32]. PLGA microparticles were modified by adsorption, incorporation, copolymerization or covalent bonding [33]. In this experiment, surface modification of PLGA microparticles was performed by physical adsorption, and the effects of chitosan modified on the size, zeta potential, and loading capacity were studied. The results are shown in Figure 1E.

From the view of the particle size, the chitosan–PLGA microparticles were larger due to the PLGA microparticles embedded during modification. This was consistent with previous research [34], and could be confirmed with a nanoparticle size analyzer and TEM (Figure 2A,B). Compared with the PLGA microparticles, a significant increase in the loading capacity of the microparticles was achieved by chitosan modification. This was mainly because chitosan was encapsulated on the surface of PLGA microparticles due to its film-forming property, viscosity and colloidal stability [35]. However, zeta potential is the main index determining the stability of an emulsion or solution. The zeta potential (Figure 3) of the PLGA microparticles was −8.47 mV, but the chitosan–PLGA microparticles had a significantly positive value of 5.20 mV. The potential changed from negative to positive, which was consistent with previous studies [36,37]. The positive potential of the chitosan–PLGA microparticles was due to the abundant amino groups on their surface. It was thus verified that the PLGA microparticles were successfully modified with chitosan.

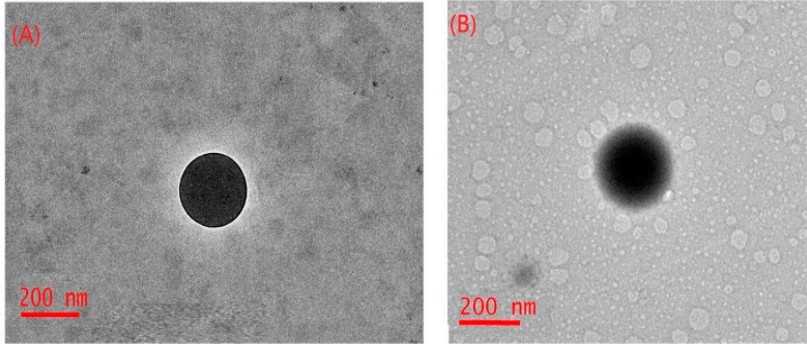

**Figure 2.** Transmission electron microscopy (TEM) images: (**A**,**B**) are poly (lactic-co-glycolic acid) (PLGA) microparticle and chitosan–PLGA microparticle, respectively.

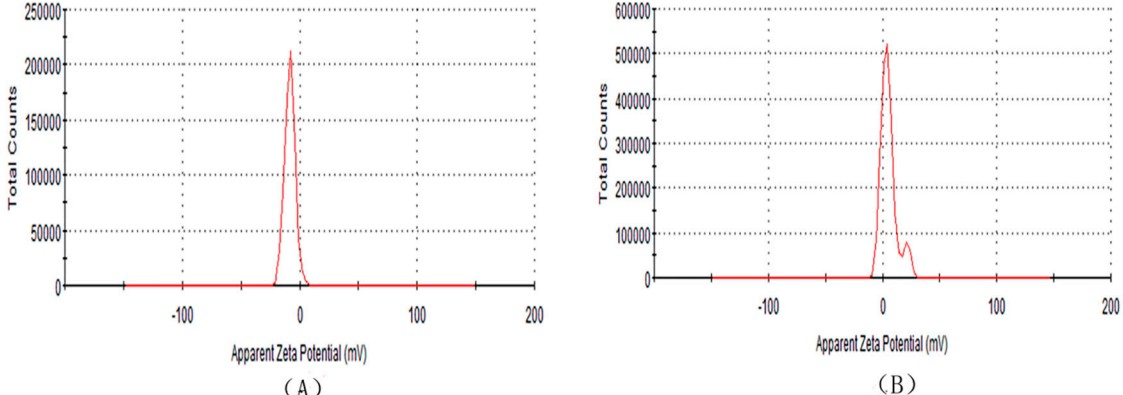

**Figure 3.** Zeta potential of (**A**) PLGA microparticles and (**B**) chitosan–PLGA microparticles.

### 3.3. Fourier Transform Infrared (FTIR) Spectroscopy Analysis

FTIR spectra of PLGA, PLGA microparticles and chitosan–PLGA microparticles are shown in Figure 4. In the FTIR spectra of PLGA, the characteristic band of the PLGA assigned to 1775 cm$^{-1}$ was associated with the −COOH stretch [38]. The saturated C−H stretching vibration of the PLGA appeared at 3117 cm$^{-1}$. In the FTIR spectra of PLGA microparticles and chitosan–PLGA microparticles, there was an obvious peak at 596 cm$^{-1}$, which was the COH surface bending of orange essential oil. There were more peaks at 500–1000 cm$^{-1}$ than PLGA, mainly because the orange essential oil contained olefins, and anthranilic acid. This confirmed that the orange essential oil was successfully encapsulated within the PLGA and chitosan–PLGA microparticles. However, the difference of PLGA, the saturated C−H stretching vibration at 3117 cm$^{-1}$ did not appear in the PLGA microparticles and chitosan–PLGA microparticles, and the other samples had a broad peak at 3421 cm$^{-1}$. This was due to the hydrogen bond formed between the −NH$_2$ of chitosan and the −OH of essential oil. It contained the C−H bond at 3421 cm$^{-1}$, and was caused by the blue shift of the C−H bond [39].

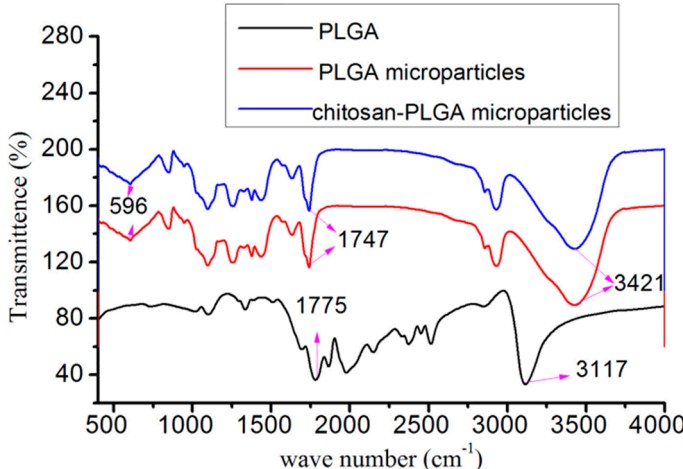

**Figure 4.** Fourier transform infrared (FTIR) spectra of PLGA, PLGA microparticles, and chitosan–PLGA microparticles in the range 400–4000 cm$^{-1}$.

### 3.4. Thermo-Gravimetric Analysis of Microparticles

The PLGA and chitosan–PLGA microparticles were analyzed using TGA. The mass loss curves were shown in Figure 5. The thermal stability of the microparticles was investigated by the TGA. Moreover, the whole weight loss could be divided into three stages. In the first stage, in the range of 0–90 °C, it could be attributed to the loss of water from the surface of the microparticles. The second phase, from 90–230 °C showed a lower weight loss. This was mainly because the orange essential oil containing olefins was vulnerable to the effects of temperature and volatilization. This might indicate that the orange essential oil was successfully embedded within the PLGA and chitosan–PLGA microparticles. In the third phase, from 250−450 °C, there was rapid quality loss. It was mainly the quality loss of chitosan and PLGA, and due to the quality of the wall, higher temperatures lead to two kinds of material fast pyrolysis. Based on the curves, it could be concluded that the thermal stability of the chitosan–PLGA microparticles was the highest. Therefore, to a certain extent, modifying the surface of the chitosan improved the thermal stability of the PLGA microparticles.

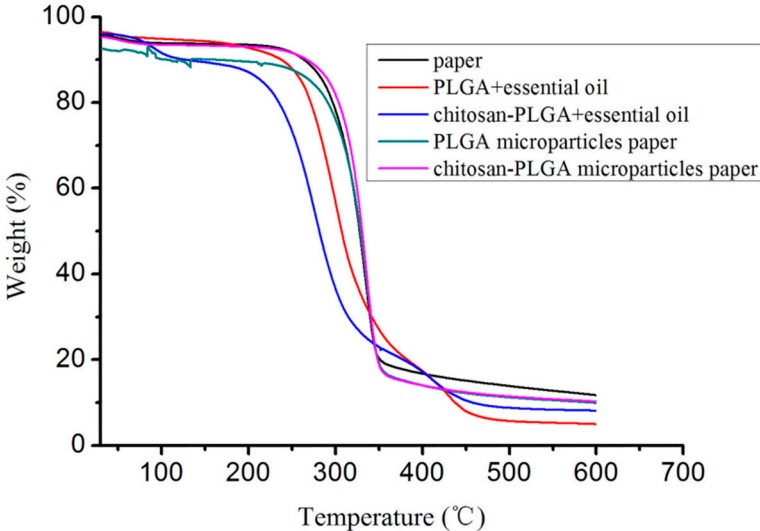

**Figure 5.** Thermogravimetric analysis (TGA) curves of untreated paper, PLGA + essential oil, chitosan–PLGA + essential oil, PLGA + essential oil paper, and chitosan–PLGA + essential oil paper.

### 3.5. Adhesive Capacity of Functional Paper

The adsorption capacity is an important index of the performance of the functional paper. First, under the same conditions, it was clear that there were more microparticles than unmodified PLGA microparticles on the surfaces of paper fibers based on the SEM results (Figure 6). Quantitative analysis of the adsorption capacity was achieved by combining formula (3) and the TGA curves (Figure 5), giving adsorption capacities of 76.32 and 324.84 mg/g, respectively. Based on the SEM results, there was a clear difference between the PLGA and chitosan–PLGA microparticles, which might be due to the surface charges of the microparticles, resulting from the mutual attraction of positive and negative charges [40]. The surface charge of PLGA microparticles was −8.47 mV, while that of the chitosan–PLGA microparticles was +5.20 mV. Furthermore, the surface of the paper fibers contained many hydroxyl and carboxyl groups, making it negatively charged. Thus, the positively charged chitosan–PLGA microparticles could adsorb better to the paper fibers.

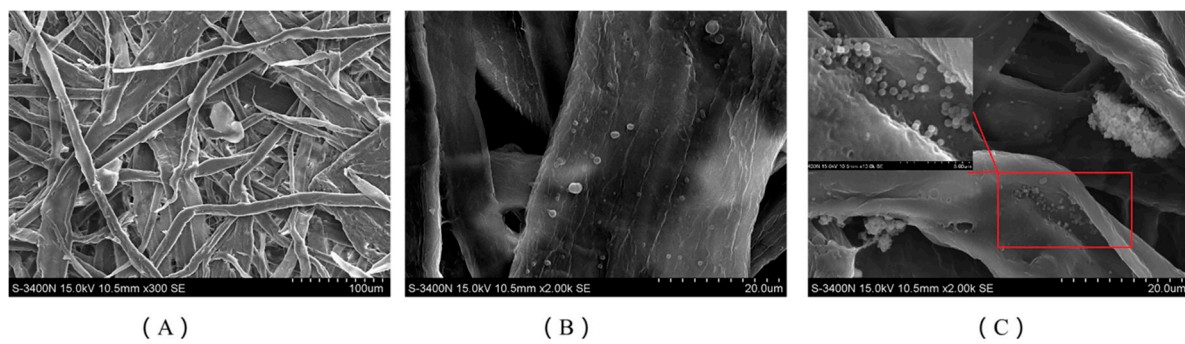

**Figure 6.** Scanning electron microscopy (SEM): (**A**) Untreated paper; (**B**) PLGA microparticles functional paper; (**C**) chitosan–PLGA microparticles functional paper.

### 3.6. Investigation of Microparticles Functional Paper Aroma Quality and Sustained-Release Effect

The aroma quality and sustained-release effect of functional paper were studied by GC-MS and the ultrafast GC Electronic Nose Heracles II. The results were as follows.

The effect of the sweet orange oil was analyzed and evaluated by GC-MS in three samples: untreated paper, PLGA microparticles functional paper, and chitosan–PLGA microparticles functional paper. The main characteristic aroma substances of the orange essential oil were determined from NIST spectra combined with related literature [41,42]. There were 72, 52 and 59 aroma components in the three samples as detected separately by GC-MS. The main aroma components of the orange essential oil were ethyl acetate, limonene, $\alpha$-pinene, myrcene, linalool, and citral. The same sample was determined at different periods using the ultrafast GC Electronic Nose Heracles II. This equipment has two advantages. Firstly, it is equipped with fast gas chromatography with a built-in pre-concentration adsorption trap, which can achieve a very low detection threshold for volatile compounds. Secondly, it has two metal capillary chromatography columns of different types, which can effectively separate and identify sweet orange oil components. The radar diagrams are shown in Figure 7. The peak areas of each component of orange essential oil were calculated using n-alkane and retention indices, and the sustained-release time of the microparticles was deduced by calculating the change in peak area over different time periods. The larger the area of the radar chart, the higher the concentration of the aroma component, and the higher the peak area.

Based on the results from GC-MS and the ultrafast GC Electronic Nose Heracles II, the encapsulation and slow-release effect were improved by chitosan surface modification. However, the relative contents of each component changed, mainly because while removing the volatile solvent ethyl acetate, terpenes and aldehydes with low boiling points in the oil were oxidized and volatilized, so that their relative contents were reduced or completely lost. The encapsulation efficiency of essential oil was 93.23% in the PLGA microcapsule. From the result of encapsulation efficiency, there was 93.23% of the essential

oil embedded in the microparticles. Furthermore, quantitative and qualitative analysis by GC-MS and the ultrafast GC Electronic Nose Heracles II showed that the composition of the essential oil did not change.

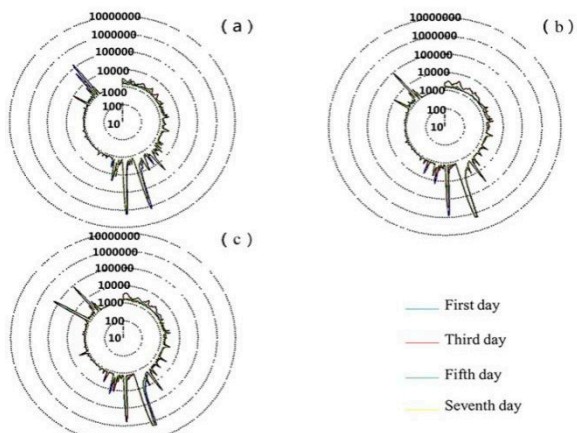

**Figure 7.** Radar diagrams of: (**a**) Untreated paper; (**b**) functional paper soaked in PLGA microparticles; (**c**) functional paper soaked in chitosan–PLGA microparticles.

## 4. Conclusions

Chitosan-modified PLGA microparticles exhibit excellent loading capacity and adsorption performance. Compared with the PLGA microparticles, the loading capacity of the chitosan–PLGA microparticles increased from 19.17% to 24.36%, the adsorption capacity increased from 76.32 to 324.84 mg/g. The process of microencapsulation had little effect on the aroma quality of the orange essential oil and could effectively prolong the sustained-release effect. Therefore, the microparticles prepared with PLGA are potentially extremely useful in the development and application of microparticle systems in the field of packing materials.

**Author Contributions:** Data curation: Z.X. and X.K.; formal analysis: Y.N.; investigation: S.W.; project administration: Z.X.; supervision: X.K.; writing—original draft: S.W.; writing—review & editing: X.K.

**Funding:** This research was funded by supported by the National Key Research and Development Program Nanotechnology Specific Project [2016YFA0200304], the National Natural Science Foundation of China [21776178], the Shanghai Engineering Technology Research Center of Fragrance and Flavour [15DZ2280100], the Shanghai Pujiang Program [18PJD048] and the Shanghai Gaofeng & Gaoyuan Project for University Academic Program Development.

**Acknowledgments:** The authors thank Junhua Liu of Shanghai Institute of Technology for his technical support for the ultrafast GC Electronic Nose Heracles II. Without his support, this work may not have been completed smoothly.

**Conflicts of Interest:** The authors declare no conflict of interest.

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
