# Peer review of "Effect of Preparation Parameters on Microparticles with High Loading Capacity and Adsorption Property Adsorbed on Functional Paper"

_coatings, doi:10.3390/coatings9110704_

Round 1

Reviewer 1 Report

Review of MSNo. 612661

Effect of preparation parameters on the application of well performance nanoparticles on functional Paper

By: Xiao et al.

General Comments:

This manuscript is about the impregnation of paper with microparticles adsorbing orange essential oil. The title of the manuscript should be changed, especially to include the meaning of the word “well performance”. The size of the particles showed here is greater than100 nm and thus the authors cannot talk about nanoparticles. Besides, the stability of the emulsion is low, since the absolute zeta potential should be larger than 20. On the other hand there are many results requiring clarification in the section of FTIR, and in the conclusion the authors claim that the functional paper shows antimicrobial effect when they did not conduct any test about it. This part is essential because the paper is considered antimicrobial. In addition, It should be considered that some volatile components of the essential oil were lost due to evaporation of ethyl acetate used as solvent, and the antimicrobial effect of the essential oil might be reduced. The use of English should be improved. Therefore, this manuscript requires major corrections before being accepted.

Specific Comments.

Title: Capital letters after the initial letter should be removed. The title is not clear especially with the word “well”. Please clarify

Summary:

Review of MSNo. 612661

Effect of preparation parameters on the application of well performance nanoparticles on functional Paper

By: Xiao et al.

General Comments:

This manuscript is about the impregnation of paper with microparticles adsorbing orange essential oil. The title of the manuscript should be changed, especially to include the meaning of the word “well performance”. The size of the particles showed here is greater than100 nm and thus the authors cannot talk about nanoparticles. Besides, the stability of the emulsion is low, since the absolute zeta potential should be larger than 20. On the other hand there are many results requiring clarification in the section of FTIR, and in the conclusion the authors claim that the functional paper shows antimicrobial effect when they did not conduct any test about it. This part is essential because the paper is considered antimicrobial. In addition, It should be considered that some volatile components of the essential oil were lost due to evaporation of ethyl acetate used as solvent, and the antimicrobial effect of the essential oil might be reduced. The use of English should be improved. Therefore, this manuscript requires major corrections before being accepted.

Specific Comments.

Title: Capital letters after the initial letter should be removed. The title is not clear especially with the word “well”. Please clarify

Summary

L. 17 The abbreviation PLGA needs to be spelled out

L. 19. This is the first mention of orange essential oil which to me is out of context. Please clarify.

L. 44. It is generally agreed that nanoparticles are those with size <100 nm, and the size written here is higher. Please clarify.

L. 78. How was the flow rate of organic phase delivered?, and also, how was the agitation rate produced and controlled? How much essential oil was lost during this process? Please clarify.

L. 105-106. Were the different suspensions mixed with KBr to form a single sheet, or each one was mixed with the KBr? Please clarify.

L. 112. What was the wavelength used to build the essential oil standard curve, and what was the meaning of “Y” and “x” of the regression equation? Please clarify.

L. 117. The authors mention ultrasonic extraction, but they do not give any data as to how it was performed. Please clarify.

L. 125. Can the authors explain the type of paper used to obtain the functional paper? Please clarify.

Eq. 2, L. 137. Can the authors clarify what is meant by the quality loss rate (W1) of untreated paper?

L. 142-149. Correct these lines please, and add the statistical analysis used.

Figure 1. The labels of both axes of all figures (1a to 1e) are not clear and should be enlarged. Figure 1e shows as label on the “y” axis the units of nm for loading capacity. Please correct.

L. 159. The particle sizes mentioned in this line, do not correspond to the values shown for PLGA 3:1, and 1:1, in Figure 1a. Please clarify.

L. 186-187. The authors mention that perhaps the PVA viscosity was too high and that it avoided microparticles separation upon heating. Do they have evidence of this fact? What was the agitation speed in this case? Please clarify.

Figure 2. The size of particles agitated at 800 rpm (Fig. 2b) is larger than those agitated at 500 rpm (Fig. 2a). The same happens for PLGA and chitosan-PLGA. Please clarify.

L. 231-232. It is not clear why the authors imply that the emulsion was stabilized by the use of chitosan, when the literature mentions that absolute zeta potential values should be greater than 20 for a stable emulsion. Please correct.

L. 248-249. The authors claim that an OH group from PLGA showed a stretching vibration, but the signal is not shown by PLGA. Please correct.

L. 250. There is no signal shown by chitosan at 3429 cm-1, while the signal at 3421 cm-1is observed even in the absence of chitosan. In FTIR there is not any absorption (also wrongly written in L. 254), there are only vibrations, stretchings, bendings, twistings, etc. Please correct.

L. 279. Can the authors clarify how they made the calculations to obtain the adsorption capacity, and to which materials are associated the two values shown here? Please clarify.

L. 284. It is not clear how the surface of paper fiber containing plenty of amino groups shows negative charge. Please clarify.

L. 297. I am not aware of any chemical named Valencia. Please correct.

L. 307. The words “higher concentration” are repeated. Please remove them.

L. 314. It is very difficult to believe that the characteristic aroma of orange essential oil does not change with losses of some volatile components. Please explain.

L. 323-324. Can the authors explain what substance is being loaded when comparing the loading capacity of the PLGA alone versus the PLGA-chitosan microparticles? Please clarify.

L. 327. The authors did not conduct any test about the antimicrobial capacity of the paper containing the microparticles of PLGA-chitosan-orange essential oil. Thus, this part should be deleted.

Author Response

For your comments, we have replied and explained point-by-point. Hope to solve your questions, and sincerely hope to get your guidance again.

Reviewer 2 Report

The manuscript deals with the effect of preparation parameters on the application of well performance nanoparticles on functional paper.

Title- Please revise capital letters.

The English language is poor and must be revised.

Please format units in accordance, e.g. “W” not “w”.

Please separate values from units, e.g. “100 W” not “100W”.

Introduction

The topics must be better linked.

Line 36- “sio2  nanoparticles”???Please format in accordance.

Materials and methods

Line 121- “The  formulas  for  calculating  the  loading  capacity  of  nano-particles  as  follow (Equation (1)).”???or formulae??

Line 124- “2.4. Preparation of the functional paper”???paper characteristics???

Line 126- “The  resulted functional paper was sealed in sample bottles for the release study.”??This method must be better described. Used area??volume of solution used???

Line 130- “SEM (S-3400N, Hitachi High-Technologies, Tokyo, Japan) at an accelerated voltage of 15 kV”??magnification levels used??

Results and discussion

Line 153- “In this work, PLGA with different monomer ratios was 75:25 and 50:50 as the wall materials of nanoparticles. The effects of monomers with different contents on the particle size, distribution and loading capacity of nanoparticles were compared. The results were shown in Figure. 1 (a).”???Figure 1, please add different superscript letters for significant differences. Moreover, please revise the discussion of the results in accordance.

Line 303- “The radar diagram was also shown in Figure. 7.”??Figure 7 has low quality and must be improved.

Conclusion

Line 326- “Therefore, the nanoparticles prepared with PLGA have great potential in the development and application of nanoparticles in food antibacterial packaging.”???or just food packaging??

References

Line 358- “Bioprocess and biosystems engineering”. Please format the title of each journal according to the guide for authors.

Author Response

(The authors gave the same response as above.)

Round 2

Reviewer 1 Report

Comments on MS Coatings-612661R1

The title is ok, but the work “adsorbed” is repeated twice, but I do not know a better way of stating this.

Al questions but one, about loading of microparticles with essential oil have been addressed with reasonable statements. The query not responded is not essential for this communication.  Therefore, I consider that the manuscript has been highly improved and is now up to the standards of the journal.

Reviewer 2 Report

The manuscript was improved.